# Using social networks to scale up and sustain community-based programmes to improve physical activity and diet in low-income and middle-income countries: a scoping review protocol

Nina Abrahams [ORCID] ,[1,2] Estelle V Lambert,[2] Frederick Marais,[3,4] Zoi Toumpakari,[1] Charlie Foster[1]

For numbered affiliations see end of article.

**Correspondence to**
Nina Abrahams;
ntabrahams@gmail.com

## ABSTRACT

**Introduction** The World Health Organisation endorses community-based programmes as a cost-effective, feasible and a 'best buy' in the prevention and management of non-communicable diseases (NCDs). These programmes are particularly successful when the community actively participates in its design, implementation and evaluation. However, they may be only useful insofar as they can be scaled up and sustained in some meaningful way. Social network research may serve as an important tool for determining the underlying mechanisms that contribute to this process. The aim of this planned scoping review is to map and collate literature on the role of social networks in scaling-up and sustaining community-based physical activity and diet programmes in low-income and middle-income countries.

**Methods and analysis** This scoping review protocol has been planned around the Arksey and O'Malley framework and its enhancement. Inclusion criteria are peer-reviewed articles and grey literature exploring the role of social networks in the scale-up and/or sustainability of NCD prevention community-based programmes in adult populations. Studies must have been published since 2000, in English, and be based in a low-income or middle-income country. The following databases will be used for this review: PubMed, Cochrane, Scopus, Web of Science, CINAHL, SocIndex, the International Bibliography of the Social Sciences, Google and Google Scholar. Books, conference abstracts and research focused only on children will be excluded. Two reviewers will independently select and extract eligible studies. Included publications will be thematically analysed using the Framework Approach.

**Ethics and dissemination** Ethical approval will not be sought for this review as no individual-level data or human participants will be involved. This protocol is registered on the Open Science Framework (https://doi.org/10.17605/OSF.IO/KG7TX). The findings from the review will be published in an accredited journal. The Preferred Reporting Items for Systematic Reviews and Meta-analyses extension for Scoping Reviews checklist will be used to support transparency and guide translation of the review.

### Strengths and limitations of this study

► This study will help fill a research gap of how social networks are used to scale up and sustain physical activity and diet community-based programmes in low-income and middle-income countries.
► This protocol is based on a widely used framework for scoping reviews and will make use of iterative steps to continuously improve rigour.
► Representatives of the public will have a chance to comment on and refine the study findings.
► Although comprehensive, the protocol has limitations with regard to search terms, language and databases used.
► Quality of evidence will not be evaluated in this scoping review.

## INTRODUCTION

The World Health Organisation (WHO) endorses a set of 16 programmes that are shown to be cost-effective, feasible and so 'best buys' in the prevention and management of non-communicable diseases (NCDs) at an individual and population level.[1] One of the 'best buy' programmes for both physical activity and diet are community-based awareness, educational and behavioural change programmes. These community-based programmes (CBPs) vary in practice and so do not have a set definition but refer broadly to programmes that target and engage a defined population in activities ranging widely from group-based interventions or mass media campaigns to environmental, structural or policy changes that are adapted to, set in, and ideally delivered by the community for that community.[2–9] Programmes wherein communities actively participate and lead in design, implementation and evaluation have been shown to be particularly creative, sustainable

and effective in improving health outcomes and developing longer-term confidence and capacity building within communities.[7 9–12] These programmes are not clinical or pharmaceutical-based but are more focused on awareness, education and creating a supportive environment for behaviour change, social change and community development.[1 2 11]

One could argue that CBPs are only truly useful insofar as they can be scaled up and sustained in some meaningful way. If the CBP does not reach enough people, then their effects are spread thin and the CBP has less chance of making a sustainable and significant impact.[2 3 5 13–15] Scale-up of an intervention has been defined as 'deliberate efforts to increase the impact of successfully tested health innovations so as to benefit more people and to foster policy and programme development on a lasting basis.[16] This can refer to the intervention or programme being introduced to a greater number of its target population, being adapted for other populations or becoming increasingly ratified at a national or international policy level.[14 16–19] Sustainability, a core component of scaling-up, has been defined as when 'the programme continues to be delivered and/or individual behaviour change is maintained; the programme and individual behavioural change may evolve or adapt while continuing to produce benefits for individuals/systems, after a defined period of time.[20] This refers to the political and institutional adoption of the programme beyond the initial funding or starting team.[16]

There is, however, limited research on how to effectively scale-up and sustain complex community-based NCD prevention programmes.[17 18] This dearth of literature is particularly pertinent in low-income and middle-income countries (LMICs). For example, Reis et al[21] conducted a systematic review on scaling-up physical activity interventions. The authors only identified 16 scaled up interventions in peer-reviewed literature, of which only two were based in an LMIC. While some evidence of scale-up does exist in LMICs, they tend to significantly focus on HIV/AIDs, maternal health and infectious diseases compared with NCD prevention.[22 23]

One potentially useful tool for improving the theory and practice of the scale-up and sustainability of complex CBPs is social network research (SNR). SNR is research that examines the relationship between actors (individuals or organisations) in a system.[24 25] The aim of SNR is to identify who is in a particular network, what their attributes are, and who they are connected to in the network and then to visually plot these relationship on a network graph.[25] There are various ways in which network structures can then be analysed.[26–28] For example, one can calculate the density of the network (how many ties/relationships there are between actors) where a high degree of ties to one stakeholder may indicate that they are considered a leader. Alternatively, there may be several densely tied sub-groups within a bigger network with bridging actors who connect these clusters. Once one understands how the network is functioning, one can then try and use these connections by generating strategies for

greater and more efficient network reach and sustainability.[25 29 30] Valente[29] suggests four main 'network interventions': identifying key actors, identifying and shifting the actions of subclusters at a time, stimulating peer-to-peer influence and altering the network (removing or adding actors into key network positions). Hunter et al[31] conducted a systematic review and meta-analysis of randomised controlled trials to identify the effectiveness of social network interventions over a range of health behaviours and outcomes. The pooled evidence indicates that social network interventions result in improved health outcomes and are particularly useful at reaching and retaining underserved populations. However, of the 37 included interventions only six studies were based in an LMIC, none of which took place in Africa.

### Broader research

To fill this gap, the authors (NA, CF, EVL, FM and ZT) are conducting research to understand and create theory of the various attributes of social networks that may impact or influence the successful implementation, scale-up, and sustainability of community based, NCD prevention, programmes. The research consists of this scoping review, a social network analysis of a case study CBP, and realist interviews with stakeholders of the CPB.

The CPB case study is a government-run healthy lifestyles partnership initiative in South Africa, the WesternCape on Wellness [WoW!].[9] The programme started in 2015 and aims to prevent, reduce and better manage NCDs by using peer networks and partnerships to promote healthy eating, healthy weight, as well as increase health-related physical activity, social connectedness and mental wellness. Currently, WoW! has most focused on physical activity and diet in adult populations and so for relevancy and scope manageability this will also be the focus of the planned scoping review.

### Scoping review research aim and questions

The aim of this scoping review is to map and collate literature on the role of social networks for scaling-up and sustaining NCD prevention physical activity and diet CBPs in LMICs. The findings of this review will be used to determine the current scope of research, help to identify gaps in the literature, and support the development of an initial programme theory as part of the broader research project aims.

The overarching research question is: Is there research on social networks within scale-up studies of community-based physical activity and diet programmes in low- and middle-income countries? And if so, what is the nature of the role of social networks?

The subquestions that will guide the scoping review are presented in table 1.

### METHODS AND ANALYSIS

Scoping reviews are useful for determining the broad scale and range of a body of literature and exploring key factors related to a particular concept of interest.[32 33] This is particularly useful when there is a paucity of evidence

| Table 1 | Scoping review questions |
|---|---|
| Descriptive | What is the volume of publications? |
| | What are the research designs of the publications? |
| | What is the geographical scope of the publications? |
| | Who are the publication authors? |
| Social networks | What types of networks and/or network interventions are described in the publications? What are they used for? |
| | Who is involved in the network(s)? |
| | What value, if any, do social networks bring to community-based programmes? |
| Community-based programmes | What types of community-based programmes are covered? |
| | What activities are included in the community-based programmes? Who is included? (age, sex, gender, health and economic status). |
| | Who is implementing these programmes? What settings are used for the programmes? |
| | What theories/theoretical approaches underpin the community-based programmes? |
| Scale-up and sustainability | What scale-up/sustainability theories are used in the publications? |
| | How is scale-up and/or sustainability conceptualised or operationalised? |
| Mechanisms | Are any potential mechanisms of scale-up and sustainability explored in the publications? |

in that field.[33] Scoping reviews are not intended for answering a specific question in detail but rather to gain a broad understanding of a particular topic.

This protocol has been planned around the methodological framework for scoping reviews outlined by the Arksey and O'Malley framework[34] as well as its enhancement.[33] The method and final reporting will also be based on the checklist provided by the Preferred Reporting Items for Systematic Reviews and Meta-analyses extension for Scoping Reviews (PRISMA-ScR).[35] These frameworks acknowledge that scoping reviews are an iterative process and that continuous engagement and refinement is needed to retrieve relevant publications and analyse reliably.[33] Any iterations or deviations from this protocol will be reported in the final findings' publication.

### Conceptual framework

This planned scoping review will form part of a larger research project that is framed by a realist evaluation. Realist evaluation is a theory-based evaluation strategy that aims to answer the question 'what works, how or why does this work, for whom and in what circumstances?'.[36–39] Namely, a theory about how a programme is thought to operate is generated based on known literature, this initial theory is then tested on a real-world case study and, based on the results, the programme theory is refined. The programme theory is built using the structure: resource mechanisms-context-reasoning mechanisms-outcomes.

Otherwise denoted as 'M(Resources)+C→M(Reasoning)=O' or (MCMO).[39]

In this research, the resource mechanism will be the social network structure of a public health intervention, such as, who the actors are, how many subgroups there are in the network, and whether any network interventions have been used to alter the network. The context is the intervention itself and the social and geographical environment within which it operates. The outcome will be increased scale-up and sustainability. Variables are defined as:

1. CBPs
▶ CBPs include any NCD prevention programmes or interventions that aim to change a defined population's psychological thinking, social interactions or behaviours. They exclude interventions on a clinical or pharmaceutical level.
  – A broad definition of NCD prevention will be used to include any programme that promotes health and prevents NCDs.[40] Activities should include promotion of physical activity, healthy diets or reducing sedentary behaviour.
  – CBPs may include programmes that use a defined community as a setting or programmes that include the community in the design, running and evaluation of the intervention.[7]
  – Activities and outcomes of CBPs may include, but are not limited to, mass media education and awareness campaigns, group-based education or activity classes, individual motivational interviewing or brief behavioural change counselling, changing built environments, improving access to resources, community capacity building and empowerment, and improved social cohesion.[2 10 11 41]

2. Scale-up and/or sustainability
▶ Scale-up, namely a deliberate intention of growth, must be emphasised by the authors as an outcome or variable of interest. It can be horizontal (different populations or more of the population, or increasing programme innovation) or vertical (embedded in policy) scale-up.[18]
▶ AND/OR.
▶ Sustainability must be emphasised by the authors as an outcome or variable of interest. It can include any interest in the programme and its effects continuing after initial implementation with specific interest in institutional and policy ratification.[20]

3. Social networks
▶ A broad definition of social networks will be used in line with previous systematic reviews of social network interventions.[31] Social networks encompass any social interaction or relationship between people or organisations, in vivo or online.
▶ To be included a publication must have measured or considered these relationships in some aspect of the intervention design or delivery. This includes recognising an existing social network as being important

to the success of the intervention (social network) or actively manipulating a network as part of the scale-up or implementation strategy (social network intervention).

4. Actors
► Any of the people or organisations mentioned that play a role in the CBP (such as intervention developers, implementers, community members, participants, funders, policy-makers).

5. Reasoning mechanisms
► Underlying social or psychological responses that potentially foster or inhibit the outcome of scale-up and/or sustainability (such as trust or communication patterns).

6. LMICs
► Countries that are low-income, lower-middle income or upper-middle income economies as categorised by the World Bank in 2021.[42]

### Inclusion and exclusion criteria

To be included, articles must explore network, context and outcome components. Namely they must explore the role that the structure of a social network may have to influence scaling-up and/or sustaining CBPs. At least two of these components should be highlighted in the abstract or summary of the publications to be considered for further review. Included articles can be peer-reviewed frameworks, theoretical papers, reviews or primary (quantitative, qualitative or mixed method) studies or grey literature, published in English since 2000. This start date is based on the rapid rise of CBP and scale-up literature since 2000 on the International Bibliography of the Social Sciences (IBSS) database. The cut-off date will be reflected in the final scoping review report. Inclusion population are human adults (19–64) and older adults (>65). If the publication has a wide age range, then the mean age should be >18 or more than 50% of the participants or beneficiaries should fall within the age range. Articles will be included if they have a wide age range

but have analysed adult and child populations separately. Child or school based only programmes are not included in the review at this stage as the broader research project focuses on adult population networks. In addition, only publications that include low-income and middle-income contexts or considerations in their analysis will be considered. Excluded studies include conference abstracts, protocols, books, any outputs not published in English and publications only focused on high-income countries (table 2).

Considering the expected dearth of literature, the planned scoping review will be kept broad. However, as this review follows an iterative process, inclusion and exclusion criteria may be further refined after the initial retrieval and review of abstracts to closer reflect the larger research project aims.[33] For example, CBPs may be further refined to only include multicomponent (complex) programmes. Namely, programmes with more than one activity or component.[3 37] In addition, articles may be refined to only include CBPs wherein the community has been intimately involved in developing and implementing the intervention.[7 10] Any adaptions to the inclusion criteria will be reported in the final publication.

### Search strategy

The search strategy will cover terms related to the three main components in this study: social networks, NCD prevention CBPs and scale-up/sustainability. Relevant search terms have been developed through review of the literature, author discussion and in consultation with the team librarian. This helps to better ensure that the review can capture the scope of literature that may use different, but related, terms. Online supplemental additional file 1 presents the search terms to be used in the scoping review.

The following databases will be used for this review: PubMed, Cochrane, Scopus, Web of Science, CINAHL, SocIndex, and IBSS. A complementary search will also be conducted through specialised social network journals: Social Networks, Applied Network Science and Journal

| Table 2 | Inclusion and exclusion criteria | |
|---|---|
| **Inclusion criteria** | **Exclusion criteria** |
| Articles exploring the role of social networks in the scale-up and/or sustainability of NCD prevention community-based programmes | Books, book chapters, book reviews |
| Peer-reviewed articles, including original research, reviews, commentaries and opinion pieces | Conference proceedings, dissertations/theses, and abstracts |
| Grey literature (eg, institution reports, government documents) | Protocols |
| Indexed in PubMed, CINAHL, Scopus, Web of Science, Cochrane, IBSS, Google (Scholar) | Website, newspaper and social media content |
| Published from year 2000 onwards | Published before 1 January 2000 |
| Language: English | Non-English publications |
| Adult population (>18) | Children population (<18) or school based |
| Publications that include contexts from low-income and middle-income countries | Publications that only include context from high-income countries |

IBSS, International Bibliography of the Social Sciences; NCD, non-communicable disease.

of Social Structure. Grey literature will be identified by reviewing the first three pages of Google and Google Scholar as well as through the IBSS database. Articles that are handsearched through the references of included publications or suggested through expert opinion will also be included.

### Study selection

NA will carry out the searches of the electronic databases. Title and abstracts will be extracted into the reference manager, EndNote, in order to remove all duplicates. NA will conduct the first stage of screening (titles only) and CF will review the first 10% of titles to test for any reviewer discrepancies. For the second (title and abstracts) and third stage of screening (full text), 50% of the publications will be reviewed by two reviewers—NA and a second reviewer who is independent of the broader research project. This will be done using the Rayyan systematic review platform, a web-based tool for systematic review management.[43] If there are no major conflicts then the remaining publications will be reviewed by NA only; however, if there are major conflicts then all publications will be independently reviewed by two reviewers. Considering the broad scope of expected studies, continuous engagement around discrepancies will be performed to improve reliability and ensure data are not missed.[44 45] Any discrepancies that cannot be agreed on will be discussed with a third reviewer (CF, EVL, FM or ZT).

### Data extraction and analysis

After full text review, included articles will be thematically analysed for their network, CBP, and scale-up properties using the Framework Approach.[44] The Framework Approach is a deductive qualitative analysis strategy that seeks to analyse data using preset categories. This structured approach is useful in answering specific questions from a diverse body of literature where one can identify themes from the outset while still being flexible and true to the data. There are five stages in the framework approach[44]:

### Data extraction

► Familiarisation with the data: The first reviewer (NA) will familiarise themselves with the data set, noting any recurrent themes and gaining an overall picture of included publication.
► Identifying a thematic framework: The framework with which to analyse and compare the data set will be based on the MCMO configuration of realist evaluation.[39 46] Two reviewers (NA and the independent reviewer) will extract the data using a customised data extraction form (table 3) to guide the analysis. The data extraction form is based on the scoping review questions. It will be independently piloted by both reviewers on a subset of publications, and refined, to improve reliability.[33 45]
► Indexing: Study characteristics to be considered are the author, year of publication, journal/publisher,

**Table 3** Planned data extraction form

| Theme | Characteristics |
| --- | --- |
| Publication details | Study ID, author, year published, journal/publisher, country, type of article (theoretical vs primary) |
| Context | Description of CBP, no of components/activities/outcomes, description of community involvement, pertinent contextual factors, CBP theories used |
| Resources | Description of the network structure/intervention, list of actors |
| Outcomes | Description of the CBP outcomes, description of the scale-up outcomes, vertical or horizontal scale-up, description of sustainability outcomes, theory use, notes on effectiveness |
| Mechanisms | Possible underlying mechanisms, theory used |

CBP, community-based programme.

type of article, the network and actors (resource), the intervention (context), possible mechanisms (reasoning) and the definitions of scale-up and/or sustainability (outcome). These will be extracted onto an Excel spreadsheet. Additional themes that emerge from the data, relevant to the MCMO framework, may also be considered. Studies will be coded and indexed based on these preset themes identified. A third reviewer will help to resolve any disagreements.

### Data analysis

► Charting: The first reviewer (NA) will collate themes and articles into a final chart in order to easily compare the themes across studies.
► Mapping and interpretation: The chart and its themes will be rearranged and thematically analysed based on MCMO configurations. Multiple theories based on possible network, programme, underlying mechanisms and scale-up/sustainability configurations will be collated to develop an initial programme theory.[38] This theory will be tested and refined as part of the broader research aims.

### Limitations

There are various limitations in this review. Scoping reviews do not provide a critique of the included publications' methodology and so cannot make claims about validity or effectiveness.[33] Publications in a language other than English are not included as this is beyond the timeline and scope of this review. In addition, the wide range of potential terms used in the literature, such as 'scale-up' versus 'dissemination' versus 'implementation'[18] 'scalability' and 'spread'[12] and the limitation on databases searched may mean that publications are missed. To mitigate these limitations, there will be continuous engagement with the literature and among authors to refine the terms and multiple reviewers will be used to analyse the publications to increase reliability and credibility of the research.[33 45] Considering scoping reviews are an iterative process, any revisions to this protocol will be reported to maintain transparency.[45] Limitations and suggestions for further research will be indicated in the dissemination of findings.

## Patient and public involvement

No patients or public will be included in formulating or conducting this research as this is a review of already published literature. However, through semi-structured interviews, as part of the broader research project, community and government groups in a middle-income country (South Africa) will receive a chance to comment on the findings and add to and refine the research at a later date.

## Ethics and dissemination

Ethical approval will not be sought for this review as no individual-level data or human participants will be involved. This protocol was registered in July 2021 on the Open Science Framework (https://doi.org/10.17605/OSF.IO/KG7TX). The findings from the review will be published in an accredited journal. The PRISMA-ScR checklist will be used to support transparency and guide translation of the review. The findings will also be used to inform the next stages in the broader research project. Findings from this broader research will also be published in a peer-review journal and shared on relevant social media platforms.

**Author affiliations**
¹Centre for Exercise, Nutrition, and Health Sciences, University of Bristol, Bristol, UK
²Health Through Physical Activity Lifestyle and Sport (HPALS) Research Centre, Faculty of Health Sciences, University of Cape Town, Cape Town, South Africa
³Department of Health, Western Cape Government, Cape Town, South Africa
⁴Africa Unit for Transdisciplinary Health Research, North-West University, Potchefstroom, South Africa

**Contributors** All authors conceptualised the review. NA led the writing of the protocol. CF, EVL, FM and ZT provided feedback and improvements. All authors read and approved the final manuscript. Acknowledgement of contribution for this protocol should also be given the University of Cape Town librarian team.

**Funding** NA is supported through a University of Bristol QR GCRF grant. Bristol's QR GCRF strategy funding is awarded to the University by Research England. Additional funding was provided by philanthropic donations from the University of Bristol's Alumni and Friends.

**Disclaimer** The funders do not have any involvement in data collection, data analysis, or data interpretation.

**Competing interests** None declared.

**Patient consent for publication** Not required.

**Provenance and peer review** Not commissioned; externally peer reviewed.

**ORCID iD**
Nina Abrahams http://orcid.org/0000-0001-5799-5780

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
