## [Reviewer comments · BMJ Open]

ARTICLE DETAILS

TITLE (PROVISIONAL)	Using social networks to scale-up and sustain community-based programmes to improve physical activity and diet in low- and middle-income countries: a scoping review protocol.
AUTHORS	Abrahams, Nina; Lambert, Estelle; Marais, Frederick; Toumpakari, Zoi; Foster, Charlie

VERSION 1 – REVIEW

REVIEWER	Sant Fruchtman, Carmen Schweizerisches Tropen- und Public Health-Institut
REVIEW RETURNED	28-Jun-2021

GENERAL COMMENTS	Thanks for this interesting protocol. It will be an exciting research project. I've added my comments and thoughts in the attached document highlighted in yellow – contact the publisher for this file. 2 main comments: - It is unclear how this study fits in a wider research project. It would be good to add some further details- The focus on NCDs doesn't seem well argued and sometimes a bit unclear. It could use some elaboration Otherwise I like the use of the frameworks and approach suggested.
--

REVIEWER	Ben Charif, Ali Universite Laval
REVIEW RETURNED	09-Jul-2021

GENERAL COMMENTS	Thank you for this opportunity to read and comment on this manuscript. I have used the Preferred Reporting Items for Systematic reviews and Meta-Analyses for systematic review protocols (PRISMA-P) checklist to structure my review. Authors sought to “map and collate literature on the role of social networks in scaling-up and sustaining community-based programmes in low- and middle-income countries.” Overall, the manuscript tackled important issues, which will be of interest to the readers of BMJ Open. Here are some suggestions to improve the manuscript. Title: • Do you make difference between "scoping review" versus "systematic scoping review?" To my knowledge, scoping reviews are defined as a type of knowledge synthesis that follows a systematic approach to map evidence on a topic and identify main concepts, theories, sources, and knowledge gaps. So authors could remove the word “systematic” from the title.
---

Abstract:

- The method subsection could include 70% of the abstract content. Please state the study design (i.e., scoping review) as a method and provide information related to the inclusion criteria.
- The following sentence is unclear: "Two reviewers will screen and analyze the publications." It could be "two reviewers will independently select and extract eligible studies."
- The following sentence could be removed/moved from the introduction subsection: "The findings will be used to determine the scope of research, identify gaps in the literature, and contribute to a broader research investigation."
- If registered, provide the name of the registry, registration number, and date.

Background:

- Please replace "intervention" with "CBP" in the following sentence: "If the intervention does not reach enough people, then their effects are spread thin and the CBP has less chance of making a sustainable and significant impact." Please note that intervention could be: the CBP (i.e., the object to be scaled up) or a scale-up strategy (i.e., the strategy used to scale up the CBP).
- Please note that there is a difference between "scalability" (Milat et al. 2012) and "scale up." Authors provided a definition of scalability in reference to the process of scaling up. Here is some content from a previously published article: "The World Health Organization defines the process of "scaling up" as "deliberate efforts to increase the impact of successfully tested health innovations so as to benefit more people and to foster policy and program development on a lasting basis" [8]. Here, we understand scaling up to optimize equity, for example, as enlarging the scope of evidence-based innovation (EBI) not only numerically but also in other respects, e.g., increasing its range to a wider variety of socio-economic backgrounds [5]. While anticipating scaling up should begin at the earliest stages of research [5, 9], we summarized the main scaling-up steps as follows [1]: (1) scalability assessment, (2) development of a scaling-up strategy, (3) implementation and evaluation of the strategy, and (4) promoting the long-term sustained use of the successfully scaled-up EBI. The scalability assessment, the preliminary and essential step in scaling up an EBI, refers to assessment of the "ability of a health innovation shown to be efficacious on a small scale and/or under controlled conditions to be expanded under real world conditions to reach a greater proportion [or range] of the eligible population, while retaining effectiveness" [5, 10]."

(<https://systematicreviewsjournal.biomedcentral.com/articles/10.1186/s13643-021-01597-6>)

- The reference 14 should be replaced with Milat et al. 2012 (<https://pubmed.ncbi.nlm.nih.gov/22241853/>).
- Please provide a clear and comprehensive explanation to help readers understand the differences or similarities between the concepts "scale up" and "sustainability."
- Page 5, lines 10-17: I would like to suggest authors to make some changes because a knowledge synthesise showed a gap in scaling up methods in high-income country contexts (<https://implementationscience.biomedcentral.com/articles/10.1186/s13012-015-0301-6>). Indeed, there is a growing interest in scaling up health innovations around the world, especially in high-income countries. But research on scaling up is effectively taking

	place in low- and middle-income countries (LMICs), especially in context of community-based primary health care in the fight against epidemics (https://implementationscience.biomedcentral.com/articles/10.1186/s13012-017-0672-y). Today, we would like to learn from this LMICs expertise to scaling up health innovations in high-income countries (https://www.idrc.ca/fr/livres/scaling-impact-innovation-public-good).  • Please note that the reference 18 (Reis et al., 2016) does not match the citation provided: Luke DA, Stamatakis KA. Systems science methods in public health: dynamics, networks, and agents. Annual Review of Public Health. 2012;33(1):357-76. • Page 5, lines 18-20. Some authors have established the Research on Patient-Oriented Scaling up (RePOS) Network to build patient-oriented research capacity in the science and practice of scaling up and ensure that patients and the public are meaningfully and equitably engaged (https://systematicreviewsjournal.biomedcentral.com/articles/10.1186/s13643-021-01597-6). Methods:  • Please indicate if a registration of your review title exists and provide registration information, including the name of the registry (e.g., OSF), registration number, and date. • I wonder how “scale-up” could be emphasized as an outcome or variable of interest. • Authors have omitted many key terms related to the concept of “scale up” (e.g., spread, widespread, scaling, upscaling, scaling out, rolling out, scalability). Please adapt your search strategy using existing one (https://systematicreviewsjournal.biomedcentral.com/articles/10.1186/s13643-021-01597-6). • “Data extraction and analysis” subsection could be split into two subsections for easier reading: 1) Data extraction and 2) Data analysis. Please describe the methods of extracting data and any processes for obtaining and confirming data from investigators. Limitations:  • Search strategy is done in English only. • No consideration of patient and public involvement. • No sex and gender considerations. Ethics and dissemination:  • Authors could register the review on the Open science framework (OSF). Reference:  • Please check and update all reference and citations. Finally, I acknowledge authors' effort for this relevant paper.
--	---

VERSION 1 – AUTHOR RESPONSE

	Reviewer 1 comments	
6	The abstract doesn't mention NCDs as the focus of your study.	Updated so that NCD and physical activity/diet focus stated in abstract

7	I would suggest reviewing the second part of the question. It sounds a bit vague to me. What is the role of what? and which function are you referring to?	Second part of question clarified.
8	It could be interesting to also include a question on who is authoring the publications? Given that your research focuses on LMICs, it could be interesting to also understand what kind of authors are publishing (HIC/LMIC, men/women,...)	"Who are the publication authors?" included as a scoping review question (table 1).
9	I would also like to understand what these networks are used for?	Question included as a scoping review question (table 1).
10	It could be good to also ask who is implementing these programmes? Government, NGOs, ..?	Question included as a scoping review question (table 1).
11	What is considered scale-up? Nation-wide, extending to more programmes?	A broad definition of scale-up has already been included under the introduction and conceptual framework headings. By remaining broad this allows us to investigate how publications have defined this concept.
12	It would be interesting to know more about it. Can you briefly describe the bigger study? Where is it? What will be done further than using SNA?	Paragraph included that explains the broader research project and how this connects to only including NCD focused CBP
13	Will this include only NCD prevention programmes, other community-based programmes will not be included? This wasn't very clear until here in the protocol. And what's the reason for this limitation? Especially given that research might be scarce, why not a wider intervention definition?	See above
14	If you keep only search in English I would also mention it as a limitation	English only presented as a study limitation.
15	What will be the role of the other authors? Given the LMIC context, it would be good to see the South African co-authors having a more prominent role?	All the authors have helped to conceptualise and create this review. The South African co-authors will lead in data collection (SNA and interviews) of the broader research project.
	Reviewer 2 comments	
16	Title: • ...authors could remove the word "systematic" from the title	"systematic" removed from title.
17	Abstract: • The method subsection could include 70% of the abstract content. Please state the study design (i.e., scoping review) as a method and provide information related to the inclusion criteria. • The following sentence is unclear: "Two reviewers will screen and analyze the publications." It could be "two reviewers will independently select and extract eligible studies." • The following sentence could be removed/moved from the introduction subsection: "The findings will be used to determine the scope of research, identify gaps in the literature, and contribute to a broader research investigation."	'Scoping review' and inclusion criteria indicated. Sentence updated. Sentence removed. OSF registration given.

	 If registered, provide the name of the registry, registration number, and date. 	
18	Background:  Please replace “intervention” with “CBP” in the following sentence: “If the intervention does not reach enough people, then their effects are spread thin and the CBP has less chance of making a sustainable and significant impact.” Please note that intervention could be: the CBP (i.e., the object to be scaled up) or a scale-up strategy (i.e., the strategy used to scale up the CBP). Please note that there is a difference between “scalability” (Milat et al. 2012) and “scale up.” Authors provided a definition of scalability in reference to the process of scaling up. The reference 14 should be replaced with Milat et al. 2012 (https://pubmed.ncbi.nlm.nih.gov/22241853/). Please provide a clear and comprehensive explanation to help readers understand the differences or similarities between the concepts “scale up” and “sustainability.” Page 5, lines 10-17: I would like to suggest authors to make some changes because a knowledge synthesise showed a gap in scaling up methods in high-income country contexts (https://implementationscience.biomedcentral.com/articles/10.1186/s13012-015-0301-6). Indeed, there is a growing interest in scaling up health innovations around the world, especially in high-income countries. But research on scaling up is effectively taking place in low- and middle-income countries (LMICs), especially in context of community-based primary health care in the fight against epidemics (https://implementationscience.biomedcentral.com/articles/10.1186/s13012-017-0672-y). Today, we would like to learn from this LMICs expertise to scaling up health innovations in high-income countries (https://www.idrc.ca/fr/livres/scaling-impact-innovation-public-good). Please note that the reference 18 (Reis et al., 2016) does not match the citation provided: Luke DA, Stamatakis KA. Systems science methods in public health: dynamics, networks, and agents. Annual Review of Public Health. 2012;33(1):357-76. Page 5, lines 18-20. Some authors have established the Research on Patient-Oriented Scaling up (RePOS) Network to build patient-oriented research capacity in the science and practice of scaling up and ensure that patients and the public are meaningfully and equitably 	“intervention” replaced with “CBP” in this sentence.  These terms have been used with similar intent (Milat et al., 2015). However, we acknowledge the importance of clear and intentional use of terms in the literature. The scale-up definition has been updated in line with the WHO ExpandNet model of scale-up (appropriate to an LMIC context). Further justification and discussion of this framework will be explored in the findings’ publication.  Reference following ‘scale-up’ definition updated to Milat et al., 2013  Explanation that sustainability is a component of scale-up given. Distinction made as included publications can explore scale-up more generally or can have just considered the sustainability component of scale-up (indicated as ‘and/or’ in search strategy)  CBP updated to “NCD prevention” in this sentence. LMIC scale-up evidence focuses heavily on primary care, infectious disease, medication/vaccine uptake but less on complex physical activity/nutrition/lifestyle. A discussion on what we can and cannot learn from other LMIC primary care etc scale-up can be brought into the discussion section of the findings’ publication. However, the dearth in literature, and the justification of focusing on LMICs for the current review still stands.  Reference has now been corrected.  This network is noted and will be considered for collaboration/dissemination of findings and, considering timeframes, possible inclusion in this review.

	engaged (https://systematicreviewsjournal.biomedcentral.com/articles/10.1186/s13643-021-01597-6).	
19	Methods:  Please indicate if a registration of your review title exists and provide registration information, including the name of the registry (e.g., OSF), registration number, and date. I wonder how “scale-up” could be emphasized as an outcome or variable of interest.  Authors have omitted many key terms related to the concept of “scale up” (e.g., spread, widespread, scaling, upscaling, scaling out, rolling out, scalability). Please adapt your search strategy using existing one (https://systematicreviewsjournal.biomedcentral.com/articles/10.1186/s13643-021-01597-6). “Data extraction and analysis” subsection could be split into two subsections for easier reading: 1) Data extraction and 2) Data analysis. Please describe the methods of extracting data and any processes for obtaining and confirming data from investigators. 	Protocol registered on OSF. Indicated under ‘ethics and dissemination’ heading and in abstract.  The authors acknowledge that this is a broad and qualitative investigation that will require continuous engagement to ensure this is done systematically. We have added “deliberate intention” as another defining factor.  Based on repeat searches with updated search terms per Charif et al (2021) on PubMed and SocINDEX, no new relevant publications for inclusion were identified. Therefore, a new search will not be run. Potential limitation of this has been acknowledged.  Section split into extraction and analysis.
20	Limitations:  Search strategy is done in English only. No consideration of patient and public involvement. No sex and gender considerations. 	 -English-only given as a limitation. -The public will have a chance to refine the findings as part of the broader research -Sex and gender indicated as a scoping review question.
21	Ethics and dissemination:  Authors could register the review on the Open science framework (OSF). 	Protocol registered on OSF.
22	Reference:  Please check and update all reference and citations 	Reference list checked and updated.

VERSION 2 – REVIEW

REVIEWER	Sant Fruchtman, Carmen Schweizerisches Tropen- und Public Health-Institut
REVIEW RETURNED	20-Aug-2021
GENERAL COMMENTS	Dear Authors, I think the protocol has improved considerably since the first submission. All my comments were addressed, thanks. Good luck with the review!